# Neuraminidase B controls neuraminidase A-dependent mucus production and evasion

**Alexandria J. Hammond**[1], **Ulrike Binsker**[1¤], **Surya D. Aggarwal**[1], **Mila Brum Ortigoza**[1,2], **Cynthia Loomis**[3], **Jeffrey N. Weiser**[1]*

**1** Department of Microbiology, New York University School of Medicine, New York, New York, United States of America, **2** Department of Medicine, Division of Infectious Diseases, New York University School of Medicine, New York, New York, United States of America, **3** Department of Pathology, New York University School of Medicine, New York, New York, United States of America

¤ Current address: Department for Biological Safety, Federal Institute for Risk Assessment, Berlin, Germany
* Jeffrey.Weiser@nyulangone.org

**Data Availability Statement:** All relevant data are within the manuscript and its Supporting Information files.

## Abstract

Binding of *Streptococcus pneumoniae* (Spn) to nasal mucus leads to entrapment and clearance via mucociliary activity during colonization. To identify Spn factors allowing for evasion of mucus binding, we used a solid-phase adherence assay with immobilized mucus of human and murine origin. Spn bound large mucus particles through interactions with carbohydrate moieties. Mutants lacking neuraminidase A (*nanA*) or neuraminidase B (*nanB*) showed increased mucus binding that correlated with diminished removal of terminal sialic acid residues on bound mucus. The non-additive activity of the two enzymes raised the question why Spn expresses two neuraminidases and suggested they function in the same pathway. Transcriptional analysis demonstrated expression of *nanA* depends on the enzymatic function of NanB. As transcription of *nanA* is increased in the presence of sialic acid, our findings suggest that sialic acid liberated from host glycoconjugates by the secreted enzyme NanB induces the expression of the cell-associated enzyme NanA. The absence of detectable mucus desialylation in the *nanA* mutant, in which NanB is still expressed, suggests that NanA is responsible for the bulk of the modification of host glycoconjugates. Thus, our studies describe a functional role for NanB in sialic acid sensing in the host. The contribution of the neuraminidases *in vivo* was then assessed in a murine model of colonization. Although mucus-binding mutants showed an early advantage, this was only observed in a competitive infection, suggesting a complex role of neuraminidases. Histologic examination of the upper respiratory tract demonstrated that Spn stimulates mucus production in a neuraminidase-dependent manner. Thus, an increase production of mucus containing secretions appears to be balanced, *in vivo*, by decreased mucus binding. We postulate that through the combined activity of its neuraminidases, Spn evades mucus binding and mucociliary clearance, which is needed to counter neuraminidase-mediated stimulation of mucus secretions.

**Funding:** This study was supported by grant from the National Institute of Health to JNW (R37 AI038446, R21 AI50867, R01 AI50893). The funders had no role in study design, data collection and analysis, decision to publish, or preparation of the manuscript.

**Competing interests:** The authors have declared that no competing interests exist.

## Author summary

*Streptococcus pneumoniae* (Spn) is a leading mucosal pathogen, whose host interaction begins with colonization of the upper respiratory tract. While there has been extensive investigation into bacterial interaction with epithelial cells, there is little understanding of bacterial-mucus interactions. Our study used mucus of human and murine origin and a murine model of colonization to study mucus associations involving Spn. The main findings reveal i) the enzymatic activity of Spn's neuraminidases (NanA and NanB) contribute to mucus evasion through removing terminal sialic acid, ii) the enzymatic activity of NanB controls expression of the main neuraminidase, NanA, and iii) Spn induces sialic acid containing mucus secretions *in vivo* in a neuraminidase-dependent manner. We postulate that during colonization, neuraminidase-dependent reduction in mucus binding enables evasion of mucociliary clearance, which is necessary to counter neuraminidase-mediated stimulation of mucus secretions. Thus, our study provides new insights into the role of Spn neuraminidases on colonization.

## Introduction

The human nasopharynx contains a diverse and extensive microbial flora, which includes *Streptococcus pneumoniae* (Spn or the pneumococcus). Spn colonization of the upper respiratory tract (URT) is typically asymptomatic. However, transit of Spn to normally sterile sites can cause a broad spectrum of disease including otitis media, pneumonia, sepsis and meningitis [1]. Regardless of the disease manifestation, successful colonization of URT is the critical first step in Spn pathogenesis.

Colonizing pneumococci are found in intimate contact with the glycocalx, the layer of mucus that coats the surfaces of the URT [2]. The major macro-molecular constituents of mucus are mucins, a heterogeneous family of heavily glycosylated proteins that form biopolymer gels through hydrophobic interactions [3]. This layer provides a barrier protecting the underlying epithelial surface from pathogens and mechanical damage. An additional characteristic of mucins are their anionic properties due in large part to extensive sialylation of their terminal glycans [4]. The association between Spn and mucus could promote its retention along the mucosal surface or, alternatively, allow for clearance through the mucociliary flow that continuously sweeps the URT. Mucus also enhances replication within the nutrient poor environment of the mucosal surface, by providing Spn a source of carbohydrates, including sialic acid, cleaved from mucin glycans [5, 6].

Although Spn resides in the mucus layer and is considered an opportunistic mucosal pathogen, there has been minimal investigation into bacterial-mucus interactions. The physical characteristics of mucus: insolubility, heterogeneity and adhesive properties, make it particularly difficult to study both *in vitro* and *in vivo*. Earlier work has shown that Spn is found associated with luminal mucus during the first few hours following acquisition, before transiting to the glycocalyx, where stable colonization of the epithelial surface occurs [2].

Spn's main virulence factor is a thick polysaccharide capsule, which inhibits adherence to immobilized mucus glycoproteins and during colonization, allowing for escape from luminal mucus [2, 7]. This corresponds with attenuated colonization of unencapsulated pneumococcal strains, demonstrating the importance of mucus interactions for colonization [2]. This effect of capsule in blocking mucus interactions could be due to mucus repulsion by capsular polysaccharides. Spn's capsule is anionic for all but a few serotypes, which could lead to charge-mediated repulsion to negatively-charged mucins. To examine whether other pneumococcal surface factors affect bacterial-mucus interactions, our lab developed a solid phase adherence

assay to look directly at the interactions between Spn and nasal secretions obtained from humans (hNF, human nasal fluid) [8]. Results from this assay demonstrated that Spn adherence to hNF was dominant among pneumococcal isolates expressing type-1 pilus. Pilus-1-dependent binding required naturally-acquired secretory immunoglobulin A (S-IgA) that binds to the type-1 pilus leading to bacterial agglutination in mucus. Recognition by specific antibody correlated with enhanced pilus-1-dependent Spn clearance in a mouse model of URT colonization when Spn were pre-treated with human S-IgA (immune exclusion). These results could explain the lower prevalence of the pilus-1 locus in isolates from adults who have accumulated naturally-acquired secretory antibody.

The purpose of this study is to determine if other pneumococcal surface factors or enzymes affect interactions with host mucus, impacting acquisition and persistence of colonization. In particular, we analyzed the effects of surface exoglycosidases, which have been previously shown to act on host glycoconjugates [9]. These cleavage events have been shown to expose epithelial cell receptors for adherence, modulate host factors involved in clearance and allow for nutrient acquisition [5, 8–10]. However, it remains unclear how these glycosidases affect mucus binding and contribute to mucus-mediated clearance.

The pneumococcal factors found to be important in adherence to mucus include the neuraminidases A and B. Remarkably, we observed that the expression of *nanA* depends on the enzymatic activity of NanB and that Spn neuraminidases promote mucus production during murine colonization. Our study highlights the importance of sialic acid removal for Spn interactions with mucus both *in vitro* and *in vivo*.

## Results

### Pneumococcal adherence to upper respiratory airway contents

To avoid the confounding effects of anti-pilus S-IgA, we used murine URT lavages (mNLs) as a source of material to look at pneumococcus-mucus interactions. URT lavages were pooled from adult mice to control for mouse to mouse variation. The association of Spn with immobilized pooled mNLs was quantified through a solid-phase adherence assay with BSA as a blocking reagent. Prior to immobilization, mNLs were sonicated for homogenization and we noticed with increased sonication time, adherence of the Type 4 (TIGR4) isolate was reduced (Fig 1A). To confirm that Spn bound to large mucus particles, we showed that prefiltering (0.45 micron) of mNLs significantly reduced binding (Fig 1B). Next, mNLs were treated with trypsin or sodium periodate to determine if Spn was binding to proteinaceous or carbohydrate structures, respectively (Fig 1C and 1D). A significant reduction in adherence was only seen with sodium periodate treatment. Lastly, to confirm that Spn was interacting with mucus particles in mNLs, we performed microscopy with alcian blue dye, which stains mucus polysaccharides (Fig 1E and 1F). Pneumococci were associated with small alcian blue staining globules (open-arrows) (Fig 1E). Spn bound to mucoid globules in mNLs were predominantly in long chains (i-iv) as compared to unbound bacteria (v) that were primarily diplococci or short chains (solid-arrows) (Fig 1E). Increased chain length, which increases surface area per bacterial particle, has been positively associated with pneumococcal adherence and colonization [11]. These experiments demonstrated that Spn binds to mucus-containing particles in the URT.

### Screening pneumococcal surface factors for their effect on mucus interactions

We focused on pneumococcal surface factors that had been shown, or hypothesized, to interact with mucus [2, 8, 9, 12–14]. Of the four surface factors (genes) initially tested using deletion

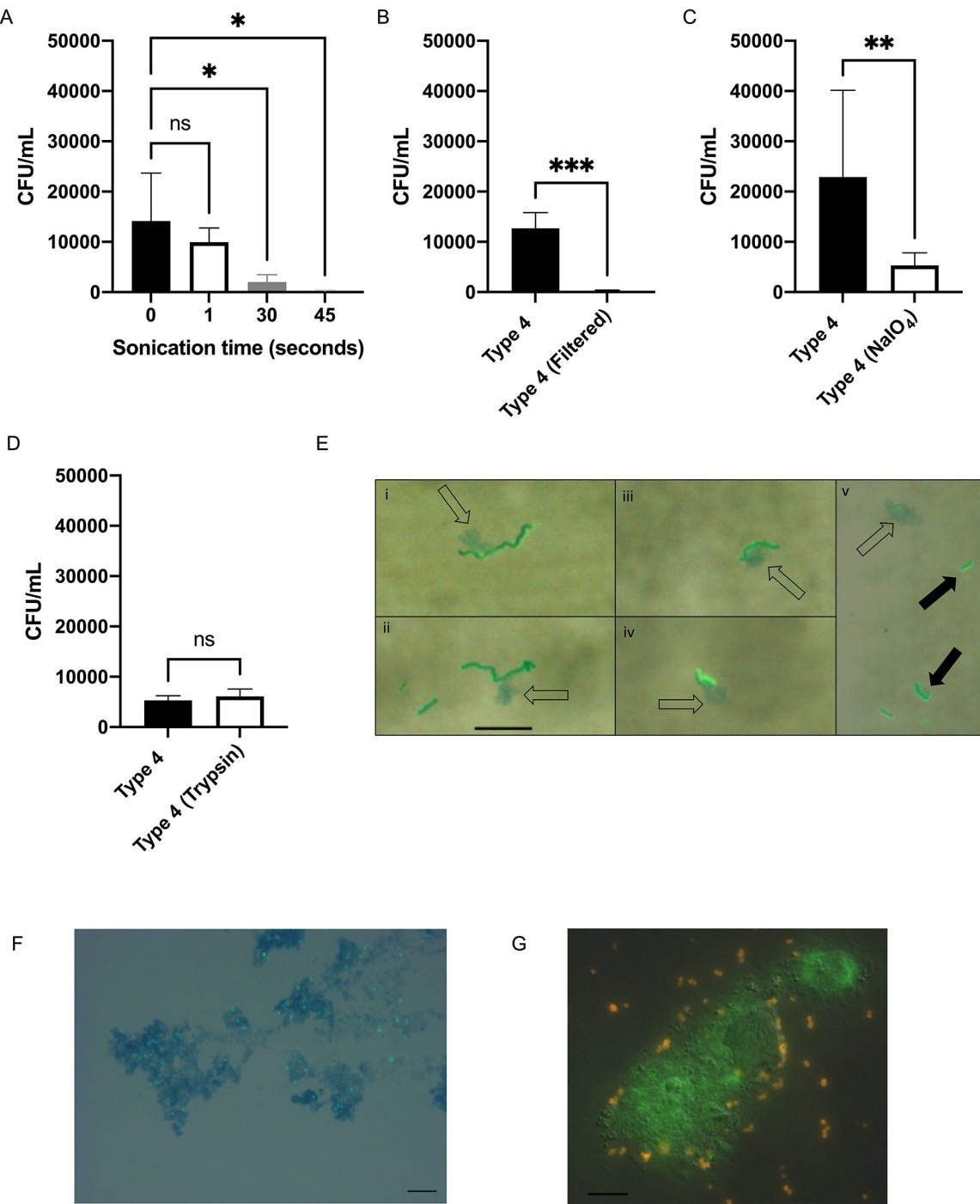

**Fig 1. Mucosal carbohydrate-mediated binding of *S. pneumoniae* to mucus particles. (A-D)** Adherence of Spn Type 4 (TIGR4) to murine nasal lavages (mNL) was analyzed in a solid-phase assay. Bacteria (2x $10^4$ CFU/100 μl DMEM) were incubated with 100μL of undiluted, immobilized, pooled mNL in presence of 0.1% BSA for 2hr at 30˚C. After 19 washes, adherent bacteria were determined by resuspending with 0.001% Triton X-100 followed by plating on TS agar plates supplemented with 200 μg/ml streptomycin. **(A)** Prior to immobilization, mNL was sonicated (Amplitude 8μM) for increasing amounts of time followed by blocking with 0.1% BSA and incubation with Spn **(B)** Filtering of mNL's with a 0.45uM filter followed by immobilization, blocking with 0.1% BSA and incubation with Spn **(C)** Treatment of immobilized mNL with 100mM $NaIO_4$ in 50mM sodium acetate buffer for 30 min at 4˚C in the dark followed by blocking with 0.1% BSA and incubation with Spn **(D)** Treatment of immobilized mNL with 50μg/mL trypsin for 30 min at 37˚C followed by the blocking with 0.1% BSA and incubation with Spn. **(E-G)** Wild-type Spn were incubated with mNL or hNF for 2hr at 37˚C and 5% $CO_2$. Scale bar represents 10μm. **(E)** Type 4 (TIGR4) incubated with mNL; (i-iv) Represents Type 4 (TIGR4) interacting with alcian blue staining material (open-arrows) (v) Type 4 (TIGR4) as diplococci or short chains not interacting with alcian blue

staining material, bacteria shown with solid arrows **(F)** Spn Type 23F incubated with human nasal fluid (hNF). Mucus (blue) was stained with alcian blue and bacteria (green) were detected using rabbit anti-capsule antibody and secondary FITC-coupled goat anti-rabbit IgG. Spn were visualized by microscopy on an Axiovert 40 CFL microscope equipped with an Axiocam IC digital camera at 100x. **(G)** Spn Type 23F incubated with human nasal fluid (hNF). Mucus (green) was stained with a Muc5AC monoclonal antibody and bacteria (orange) were detected using rabbit anti-capsule antibody. Spn were visualized by microscopy on a LM- Zeiss AxioObserver at 63x. Experiments were performed in duplicates and mean values of three independent experiments are shown with error bars corresponding to S.D. *,p<0.05; **,p<0.01;***, p<0.001 by Kruskal-Wallis test, followed by Dunn's multiple comparison test for multiple comparison (A) or an unpaired-T test (B,C, D).

mutants: capsule (*cps*), type-1-pilus (*rlrA*), choline binding protein A (*cbpA*) and mucin binding protein (*mucBP*) were tested for their contribution to adherence to mNLs, only capsule (*cps)* affected Spn's interactions with mucus in the solid phase binding assay (Table 1, Fig 2A and 2B). The increased adherence by the unencapsulated mutant was consistent with other published results using a different mucus adherence assay [2]. The lack of an effect for RlrA was expected, as mNLs lack naturally-acquired anti-pilus S-IgA (Fig 2B) [8].

Next, we tested pneumococcal surface enzymes for their role in mucus binding in the Type 4 (TIGR4 isolate) background, if an effect was detected, we also tested a knockout mutant in the Type 23F background (Table 1). We saw no effect of the O-deacetylase EstA, which acts on sialic acid residues; no effect of StrH, an exoglycosidase that cleaves terminal β1-linked N-acetylglucosamine; and a strain-specific effect with BgaA, an exoglycosidase which cleaves terminal beta-galactose and, consequently, these genes were not considered further [12, 15, 16].

## Pneumococcal neuraminidases negatively affect binding to mucus

Neuraminidases are exoglycosidases, which cleave terminal sialic acid (neuraminic acid), including on residues found within host glycoconjugates. Neuraminidases are found in all pneumococcal strains, and are encoded by three different genes *nanA*, *nanB* and *nanC*. NanA, which cleaves α2,3- α2,6- and α2,8-linked sialic acid, is present in (100%) of strains, while NanB and NanC, both with specificity for α2,3-linked sialic acid, are expressed by 96% and 51% of strains, respectively [18–20]. The function of neuraminidases has been attributed to release of sialic acid for nutrition or for promoting adherence to epithelial cell receptors which

**Table 1. Spn genes tested for their effect on mucus interactions.** Spn surface factors (row 2–5) and enzymes (row 6–10) tested for their effect on mucus interactions. Data shows the results of adherence assays of a Spn gene-knockout strain with mNF or, as noted with hNF compared to wild-type Spn. (-) indicates that there was no increase in adherence and a (+) indicates there was an increase in adherence. We tested these genes for their role in mucus binding in the Type 4 (TIGR4 isolate) background, if an effect was detected, we also tested a knockout mutant in the Type 23F background.

| SP_# | Gene Name | Gene Function | Increased adherence (mNL) |
|---|---|---|---|
| SP_0461–0468 | *rlrA* | Type 1 pilus; Adhesin | - |
| SP_2190 | *cbpA/pspC* | Choline binding protein A; Adhesion to pIGR, binds human secretory component | - |
| SP_0342–0366 | *cps* | Capsule; Capsule biosynthesis | + |
| SP_1492 | *mucBP* | Mucin binding protein | - |
| SP_0614 | *estA* | O-deacetylase; Removes acetyl groups from sialic acid facilitating neuraminidase activity | -[#] |
| SP_1693 | *nanA* | Neuraminidase; Exo-glycosidase, binds and releases terminal −2,3, −2,6 and −2,8 linked sialic acid | +[Ψ] |
| SP_1687 | *nanB* | Neuraminidase; Exo-glycosidase, binds and releases terminal α2,3 sialic acid | + |
| SP_0648 | *bgaA* | B-galactosidase; Binds and releases β1–4 galactose | -[*] |
| SP_0057 | *strH* | N-acetylhexosaminidase; Binds and releases N-acetylglucosamines | -[~] |

*Strain-dependent, strain Type 23F (Yes), TIGR4 (No)

[Ψ] Effect observed with Type 23F. No significant effect in strain TIGR4 which expresses a truncated, secreted form of the enzyme

[~]Only tested in the Type 23F background

[#] Additionally, tested in hNF due to the difference in acetylation in humans versus mice [17]

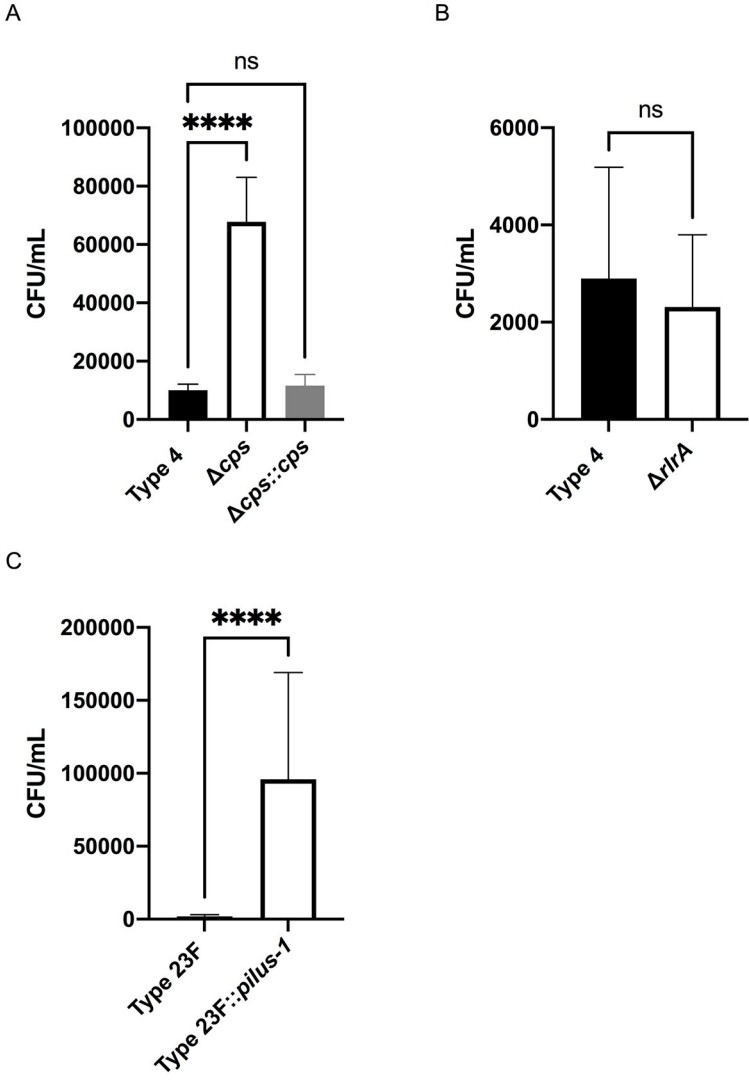

**Fig 2. Screening of pneumococcal surface factors interacting with mucus.** Adherence of WT Spn and isogenic mutants to pooled mNL or hNF was assessed in a solid phase assay. Bacteria ($2x\ 10^4$ CFU/100μl DMEM) were incubated with 100μL of undiluted, immobilized, pooled mNL in presence of 0.1% BSA for 2hr at 30˚C. After 19 washes, adherent bacteria were determined by resuspending with 0.001% Triton X-100 following plating on TS agar plates supplemented with 200 μg/ml streptomycin. **(A)** Adherence of Type 4 (TIGR4), Δcps and Δcps::cps to mNF. **(B)** Adherence of Type 4 (TIGR4) and ΔrlrA to mNF. **(C)** Adherence of Type 23F and Type 23F::pilus-1 to hNF. Experiments were performed in duplicates and mean values of three independent experiments are shown with error bars corresponding to S.D. ****,p<0.0001 by one-way ANOVA followed by Dunnett's multiple comparison test for multiple comparison (A) or Mann-Whitney test (B,C).

become exposed upon removal of sialic acid [5, 9, 10, 21]. Spn neuraminidases have also been shown to desialylate the surface of other microbes that reside in the human URT potentially providing a competitive advantage [22].

NanA, of the extensively studied Type 4 (TIGR4) isolate, contains a frameshift mutation 5' to the domain expressing the LPxTG-cell wall anchoring motif, resulting in secretion of the enzymatic portion of the protein [23]. As this mutation is an anomaly among pneumococci, we investigated the neuraminidase's role in mucus interactions using an isolate of Type 23F

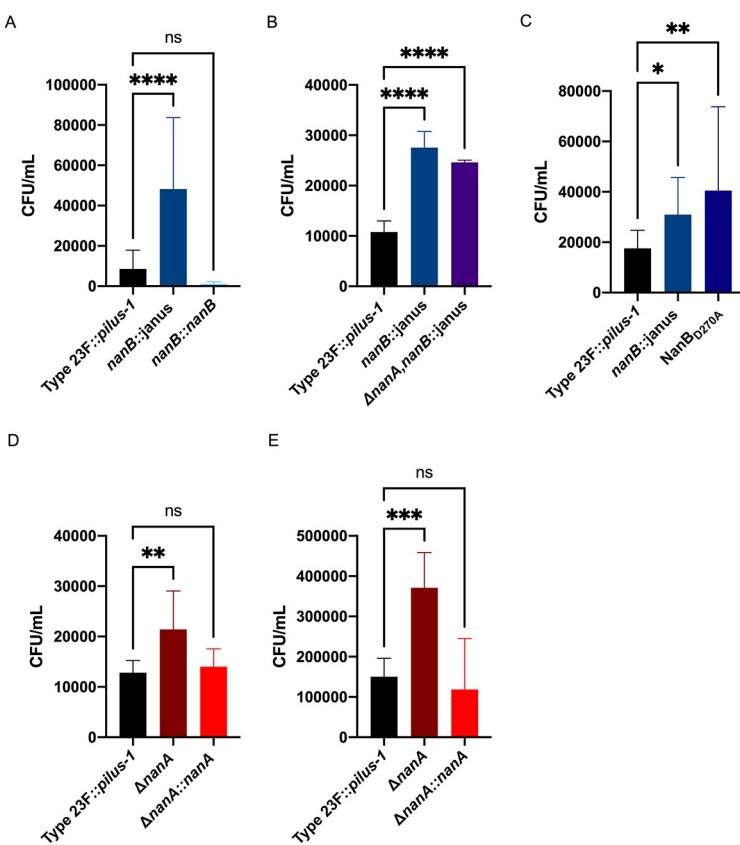

**Fig 3. NanB and NanA mediate mucus evasion.** Adherence of Type 23F::*pilus-1* and isogenic mutants to pooled mNL and hNF was assessed in a solid phase assay. Bacteria (2 x 10$^4$ CFU/ 100 µl DMEM) were incubated with 100µL of undiluted, pooled mNL or 10 µg hNF in presence of 0.1% BSA for 2hr at 30°C. After 19 washes, adherent bacteria were determined by resuspending with 0.001% Triton X-100 following plating on TS agar plates supplemented with 200 µg/ml streptomycin. **(A)** Adherence of Type 23F::*pilus-1*; *nanB*::janus and *nanB*::*nanB* to mNF **(B)** Adherence of Type 23F::*pilus-1; nanB*::janus and Δ*nanA,nanB*::janus to hNF **(C)** Adherence of Type 23F::*pilus-1*; *nanB*::janus and NanB$_{D270A}$ to hNF **(D)** Adherence of Type 23F::*pilus-1*; Δ*nanA* and Δ*nanA*::*nanA* to mNF **(E)** Adherence of Type 23F::*pilus-1*; Δ*nanA* and Δ*nanA*::*nanA* to hNF. (A) A representative data-set is shown with error bars corresponding to S.D. ****,p<0.0001 by one-way ANOVA followed by Dunnett's multiple comparison test. (B-E) Experiments were performed in duplicates and mean values of three independent experiments are shown with error bars corresponding to S.D. *,p<0.05; **,p<0.01; ***,p<0.001; ****,p<0.0001 by one-way ANOVA followed by Dunnett's multiple comparison test for multiple comparisons.

that lacks *nanC* and the type-1 pilus [18]. In addition to testing adherence in mNLs, we used hNF pooled from healthy adults as a more physiologic substrate, since Spn resides along the mucosal surface of the human URT. In order to detect robust mucus binding to hNF, we utilized a previously described Type 23F construct with an inserted type-1 pilus (Type 23F::*pilus-1*) and confirmed its robust adherence to hNF in our assay (Fig 2C) [8]. Binding of the Type 23F::*pilus-1* to large mucus particles in hNF was confirmed using alcian blue staining and a monoclonal antibody against human mucin MUC5A/C (Fig 1F and 1G). Construction of an insertion mutation in *nanB* resulted in a ~5-fold increase in adherence to mNL that was fully corrected in the chromosomally complemented strain (Fig 3A). The *nanB*::janus knockout mutant strain also displayed a ~2.5 fold increase in adherence to hNF (Fig 3B). NanB contains both an enzymatic and lectin binding domain, and to determine the role of the former in mucus adherence, we constructed a mutant with an enzymatically inactive NanB [24]. To

generate this strain, we mutated an aspartic acid residue, a conserved feature of sialidase active sites, which is an acid/base catalyst situated within a loop adjacent to the active site [24]. The enzymatically inactive strain (NanB$_{D270A}$) demonstrated significantly increased adherence to hNF compared to wild-type controls, confirming that the enzymatic activity of NanB is responsible for mucus evasion (Fig 3C). Next, we tested an unmarked, in-frame deletion of *nanA* and a chromosomally-corrected mutant to test for adherence to mNL and hNF (Fig 3D and 3E). We observed a similar increase in adherence in the *nanA*-deficient strain with both substrates (mNL and hNF) that was fully corrected in the complemented strain (Fig 3D and 3E). The increased adherence phenotype of the *nanA*- and *nanB*-deficient strains was not due to growth as longer chains, as determined through microscopy. To determine if there was an additive effect of NanA and NanB in mucus binding, we tested a double Δ*nanA*,*nanB*::janus mutant for adherence to hNF (Fig 3B). Interestingly, the double mutant showed comparable adherence levels to the *nanB* single mutant. This suggested that there was no additive effect of the two neuraminidases in mucus binding, and raised the possibility that they function in the same pathway.

## NanB regulates *nanA* in a sialic acid-dependent manner

To better understand the relationship between the two neuraminidases, we first validated the loss of enzymatic activity in our mutants using a sensitive neuraminidase activity assay. The Type 23F::*pilus-1* strain showed similar levels of cell-associated neuraminidase activity compared to other clinical isolates and as predicted this level was reduced for strain Type 4 (TIGR4) isolate that expresses a secreted version of NanA (Fig 4A). We noted high levels of neuraminidase activity for Type 23F::*pilus-1*, and the chromosomally-corrected mutants of *nanA* and background levels of activity for the deletion mutant of *nanA* (Fig 4B). Surprisingly, no appreciable levels were observed for the *nanB* deficient strain and the NanB$_{D270A}$ mutant, both of which still encode for NanA. Correction of the *nanB* mutant restored normal wild type levels of neuraminidase activity. Further, the double neuraminidase mutant also did not display any neuraminidase activity. The inactivation of either *nanA* or *nanB*, therefore, was sufficient to eliminate completely neuraminidase activity, further supporting the hypothesis/finding that these enzymes might function in the same pathway (Fig 4B).

Next, we compared transcription of the neuraminidase genes during *in vitro* growth in nutrient rich media to determine how *nanA* and *nanB*, which are expressed on separate transcriptional units in the same genetic locus, might interact [25]. *NanB* expression was as anticipated, with no difference in expression observed between Type 23F::*pilus-1*, *nanA*-deletion and NanB$_{D270A}$ mutants (Fig 4C). *NanB* was not transcribed in the *nanB*-deletion mutant and double neuraminidase mutant strains. As expected, the *nanA* transcript was not detected in the *nanA*-deletion mutant and double neuraminidase mutant, but surprisingly the *nanA* transcript was also not detected in the *nanB*-deletion mutant (Fig 4D). The NanB$_{D270A}$ mutant in which the catalytically inactive form of *nanB* is transcribed also lacked *nanA* transcripts, suggesting that the enzymatic function of NanB is necessary for *nanA* expression. As transcription of *nanA* is increased in response to the presence of free sialic acid and controlled by the positive regulator NanR, we checked expression levels of *nanR* under these growth conditions (Fig 4E) [25]. Expression of *nanR* was unaffected by the expression of functional NanA or NanB, suggesting that the enzymatic activity of NanB may regulate *nanA* through post-transcriptional effects on the NanR regulator. We then provided a host source of sialic acid to test NanB regulation of *nanA* under the conditions used in our adherence assay (Fig 4F). Neuraminidase activity in DMEM was increased by addition of mNL as the source of sialic acid and this effect was not due to increased bacterial growth. Again, neuraminidase activity was dependent on

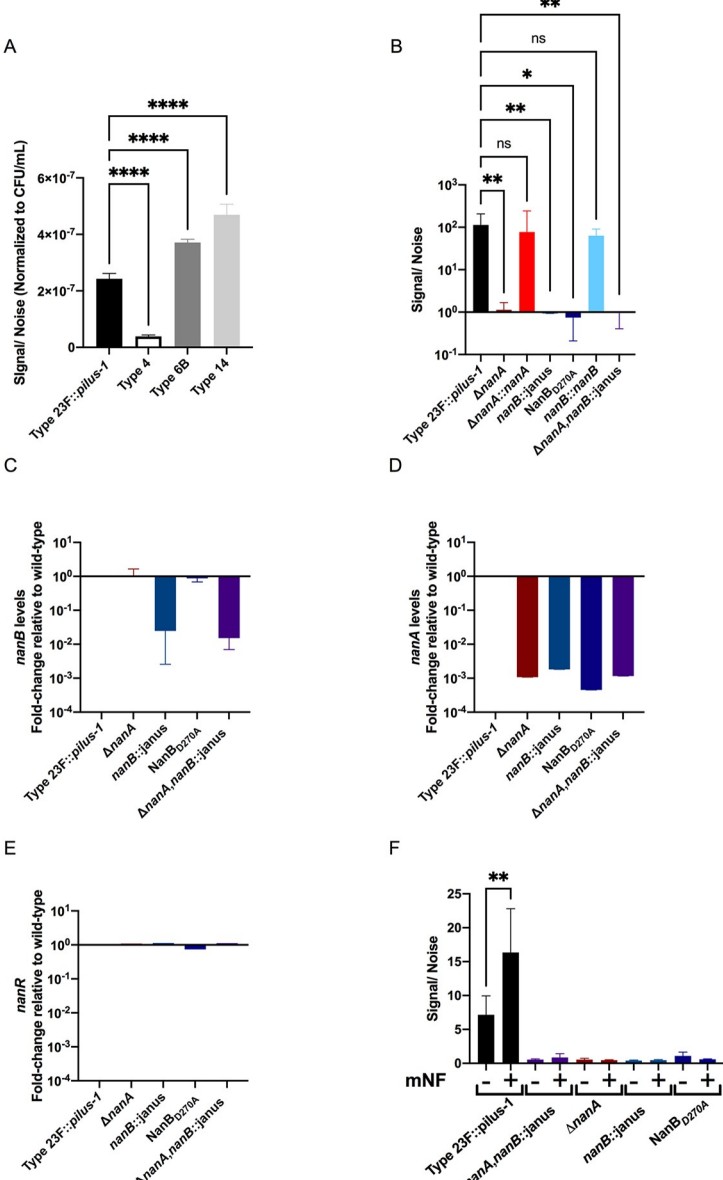

**Fig 4. NanB regulates *nanA* in a sialic acid-dependent manner. (A-B)** Neuraminidase activity by Spn was quantified using NA-STAR kit (Thermo-Fischer Scientific, USA). Pneumococcal isolates of different serotypes (**A**), or the defined isogenic mutants of Type 23F::*pilus-1* (**B**) were grown up to $1 \times 10^8$ CFU/mL, centrifuged and resuspended in PBS at a density of $10^7$ CFU/mL. Bacterial suspensions were then sonicated (Amplitude, 8 μm) and the supernatant incubated in buffer and NA-STAR substrate for 20 min at room temperature before reading the chemiluminescent signal. To analyze the data, the signal was divided by the noise (PBS control). **(A-B)** Experiments were performed in duplicate and mean values of three independent experiments are shown with error bars corresponding to S.D.**,p<0.01; ****, p<0.0001 by ordinary one-way ANOVA followed by Dunnett's multiple comparison test for multiple comparisons **(C-E)** Transcription level of *nan* genes was measured using quantitative RT-PCR. Spn strains were grown in TS at 37°C to $OD_{620} = 1.0$ followed by RNA extraction. Data shown as the fold-change was calculated relative to wild-type (Type 23F::*pilus-1*) for **(C)** *nanB* **(D)** *nanA* **(E)** *nanR*. Mean values of two independent experiments performed in duplicate are shown. **(F)** Neuraminidase levels and activity in response to sialic acid found in mNLs were also assessed using the NA-STAR kit. Type 23F::*pilus-1* and isogenic mutants were grown in TS for 1hr, spun down and resuspended in DMEM. These samples were added at a 1:50 dilution to DMEM alone, or DMEM with mNL. They were incubated at 37°C and 5% $CO_2$ for 3 hours, centrifuged and resuspended in PBS. Samples were then treated in the same manner as Fig 4B. The experiment was performed in duplicate and mean values of 3 independent experiments are shown with error bars corresponding to S.D.**,p<0.01 by an unpaired T-test to compare control to treatment.

the presence of *nanB* and the enzymatic activity of NanB. Together these observations were consistent with the use of host glycoconjugates from the URT by Spn as a source of sialic acid and that this sialic acid is liberated by NanB to trigger expression of *nanA*.

## Spn neuraminidases exert their effect on mucus through desialylation

To verify that increased mucus binding was due to the neuraminidase activity of Spn, we pre-treated immobilized hNF with *Vibrio cholerae* neuraminidase (Fig 5A). Pre-treatment with exogenous neuraminidase complemented the phenotype of the single and double *nanA* and *nanB deficient* mutants. With pre-treatment with exogenous neuraminidase these mutants no longer showed increased adherence. These results confirmed the role of the enzymatic domain of the pneumococcal neuraminidases in sialic acid removal to promote mucus evasion. There was no significant difference between the *nanB*::janus and double neuraminidase mutant, consistent with the observation that NanB is required for *nanA* expression.

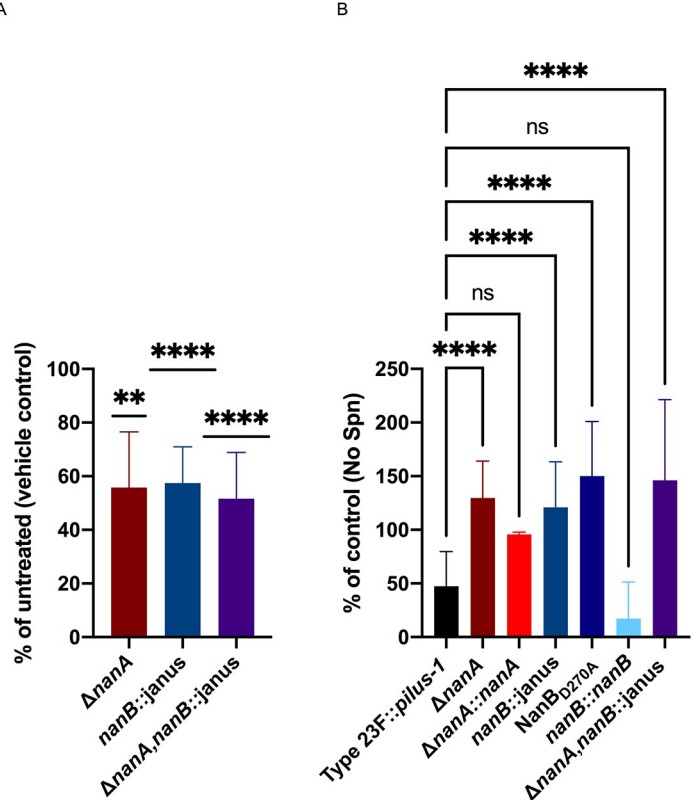

**Fig 5. NanB and NanA mediate mucus evasion through removal of sialic acid.** (A) Adherence of Type 23F::*pilus-1* and isogenic mutants to pooled human nasal fluid (hNF) was assessed in a solid phase assay. Immobilized hNF (10μg) was pre-incubated with exogenous neuraminidase from lyophilized *Vibrio cholerae* or vehicle control (calcium saline) alone. (B) The ability of Type 23F::*pilus-1* and isogenic mutants to remove sialic acid from mucus was quantified by ELISA. hNF was immobilized in a microtiter plate incubated with bacteria (1 x 10$^6$ CFU/100 μl DMEM) for 4hr at 37˚C. Binding of biotinylated lectin Mal-I to α-2,3 sialic acid was detected using peroxidase-coupled streptavidin. The values of control wells without hNF were subtracted from each measured value. Results are illustrated as % of control of untreated hNF. (A-B) Experiments were performed in duplicates and mean values of three independent experiments are shown with error bars corresponding to S.D. **,p<0.01, ***; ****, p<0.0001. A one sample T test and Wilcoxon test to a theoretical mean of 1 (A) or a one-way ANOVA followed by Dunnett's multiple comparison test for multiple comparisons were used as statistical tests (B).

Next, to quantify Spn removal of sialic acid from mucus, we developed a sensitive ELISA to detect removal of sialic acid. In this ELISA, hNF was immobilized and following incubation with bacteria, sialic acid removal was quantified through binding of a lectin specific for α2,3-linked sialic acid. We observed loss of lectin binding with Type 23F::*pilus-1*, which correlated with the presence neuraminidases (Fig 5B). In contrast, lectin binding was maintained following incubation with the single and double *nanA* and *nanB* mutants, demonstrating an inability to removal sialic acid due to the lack of neuraminidase enzymatic activity. No difference in the ability of the single mutants, double neuraminidase mutants or the enzymatically-inactive NanB$_{D270A}$ point mutant to remove α2,3-linked sialic acid was again consistent with our observation that the NanB neuraminidase activity regulates *nanA* expression. The total absence of neuraminidase activity in the *nanA*-deficient strain, when *nanB* is still expressed, also suggested that NanB does not contribute to the bulk of mucus desialylation independently of its effects on *nanA* expression.

## Effect of Spn neuraminidases in murine colonization

To determine whether the neuraminidases impact colonization fitness, we measured URT colonization density using an adult mouse model. We continued to use the Type 23F::*pilus-1* strain to correlate with our *in vitro* studies, although the presence of the pilus locus has no impact on murine colonization [8]. As a sensitive measure of the contribution of these genes *in vivo*, we looked at the competitive index of the parent strain compared to the double neuraminidase mutant, focusing on early timepoints when Spn is seen predominantly associated with luminal mucus (Fig 6A) [2]. The double neuraminidase mutant outcompeted Type 23F::*pilus-1* at 4 hr and 24 hr post-infection, although this increased retention of the double neuraminidase mutant soon after inoculation was relatively modest. At 5 days post-infection, the advantage for the double neuraminidase mutant over Type 23F::*pilus-1* was lost, indicating a temporary advantage to the increased mucus-binding phenotype of the mutant. When the strains were tested individually, no difference in colonization levels was detected at 4 hr, 24 hr or 5 days post-infection (Fig 6B). In infant mice, which are more susceptible to Spn infection, there was also no difference in colonization at 24 hr or 5 days post-challenge. As we only observed the contribution of the neuraminidases in a competitive infection, when Type 23F::*pilus-1* could be complementing the mutant; this suggested the possibility of additional effects of the neuraminidase locus.

## Spn stimulates mucus containing secretions *in vivo* in a neuraminidase-dependent manner

To more fully understand the impact of neuraminidases on mucosal interactions, we examined URT tissue sections at day 5 post-infection comparing Type 23F::*pilus*-1 and the single and double neuraminidase mutants. Similar numbers of pneumococci were seen along the mucosal surfaces, based on low-magnification images corresponding the equivalent density of Spn obtained by quantitative culture. Using a Sambucus Nigra Lectin (SNL, EBL) that detects α2,6 linked sialic acid, we observed increased levels of staining along the surface of the respiratory epithelium when infected with Type 23F::*pilus-1*, that was reduced in the double neuraminidase mutant, with the latter resembling the mock-infected group (Fig 7A, upper row). These observations were counter-intuitive since Type 23F::*pilus-1* cleaves sialic acid, but more sialic acid was detected than in mock-infected animals. This raised the possibility that infection with Type 23F::*pilus-1* was inducing more mucus containing secretions. Accordingly, a thicker glycocalx layer and increased staining in glandular structures along the mucosa in Type 23F::*pilus-1* colonized mice was seen by alcian blue-PAS staining for mucopolysaccharides relative to mock-infected controls (Fig 7A, lower row). To quantify the results, we compared sialic acid

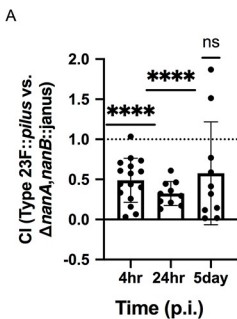

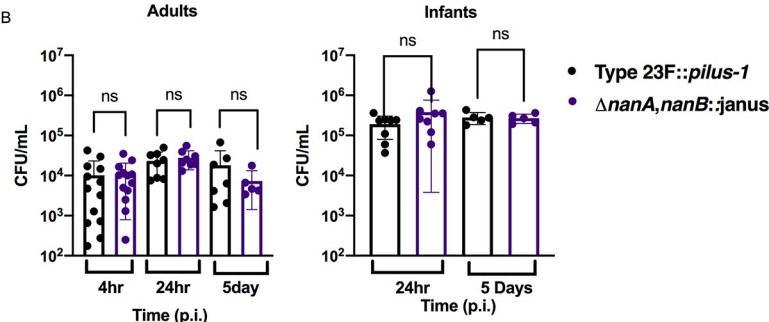

**Fig 6. Effect of neuraminidases on colonization *in vivo*. (A)** Adult mice were intranasally infected with a suspension containing equal amounts of Type 23F::*pilus-1* and isogenic double neuraminidase mutant. Colonization density was assessed 4h, 24h and 5 days p.i. in the URT lavages to calculate the competitive index (CI). Dotted line represents CI = 1. Group medians were compared to a CI = 1 by one sample T and Wilcoxon signed rank test and resulting p-values are indicated ****, p<0.0001. **(B)** Adult (left panel) and infant (right panel) mice were intranasally infected with Type 23F::*pilus-1* or Δ*nanA,nanB*::janus. Colonization density was assessed at 4h, 24h and 5 days p.i. in adult mice and 24h and 5 days p.i. in infant mice. Experiments were repeated twice and groups represent n = 5–15 animals. Error bars correspond to S.D. by Mann-Whitney test.

levels and amounts of mucus in immunoblots of mNLs using a sialic acid binding lectin and a mAb to the muc5ac mucin (Figs 7B, 7C and S1). We observed a significant increase in sialic acid- and mucin-containing secretions in Type 23F::*pilus-1* as compared to both the Δ*nanA, nanB*::janus mutant and mock-infected animals. When we compared the single neuraminidases mutants to Type 23F::*pilus-1*, we observed that in both tissues sections staining for sialic acid or mucopolysaccharides and immunoblots of mNLs to quantify sialic acid or mucin, the *nanA-* and *nanB-deficient* mutants showed a staining pattern similar to the double neuraminidase-deficient strain (Fig 7A–7C). Together, these results demonstrated that during URT colonization, Spn stimulates sialic acid containing secretions in a neuraminidase dependent-manner and that both neuraminidases are required for these effects.

## Discussion

The interactions of a mucosal pathogen with host mucus were examined in this study. Previous investigations on mucosal pathogens have focused on interactions with epithelial cells rather than either the firm and/or loose mucus layers coating the URT. We decided to focus on these interactions in particular as Spn is found embedded in the mucus layer during colonization. Mucus is especially challenging to incorporate in *in vitro* studies because of its marked heterogeneity, insoluble components and its viscous nature. Many studies have dealt with these

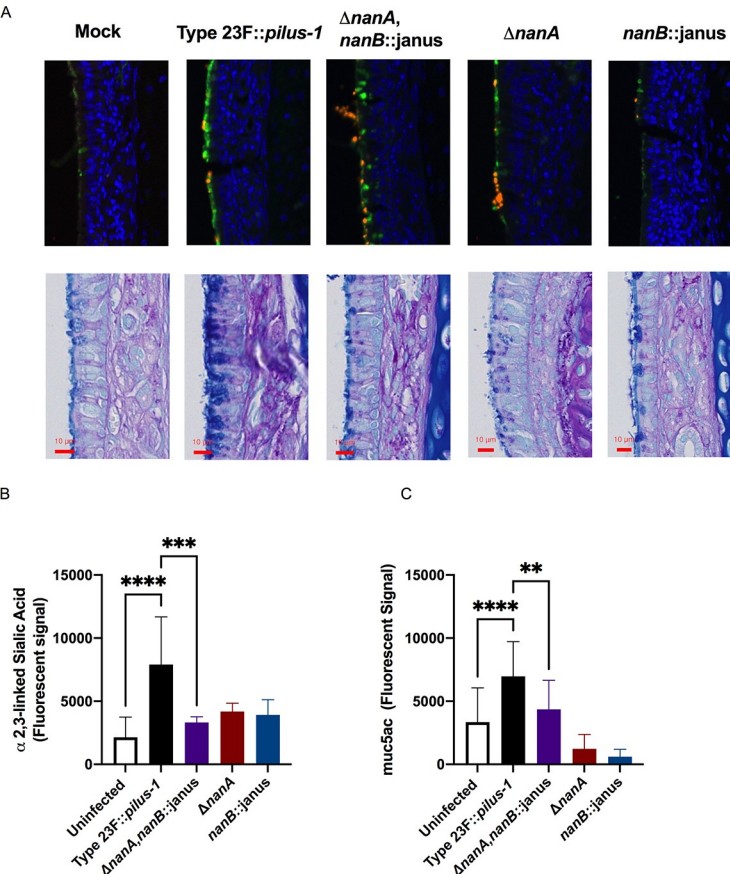

**Fig 7. Spn neuraminidases stimulate mucus containing secretions in a neuraminidases-dependent manner. (A)** URT tissue sections of mock- or Spn-infected infant mice were examined at day 5 post-infection. Sialic acid containing secretions were visualized through a SNL lectin staining that detects α-2,6 linked sialic acid (upper row), where green staining indicates sialic acid containing material and orange staining indicates Spn; or alcian blue-PAS staining for mucopolysaccharides (lower row). **(B-C)** To quantify sialic acid and mucus containing secretions in the URT of mice, retro-tracheal lavages were obtained from infant mice at day 5 p.i.. Immunoblots were performed with the lavages to determine the amount of α-2,3 linked sialic acid **(B)** or muc5ac mucin **(C)**. Experiments were performed in duplicates and mean values of three independent experiments are shown with error bars corresponding to S.D., *,p<0.05 **, p<0.01, ***,p<0.001 by 1-way ANOVA followed by Dunnett's multiple comparison test.

limitations through the use of purified or semi-purified mucins, rather than native mucus [26–28]. An additional challenge in using human mucus with human commensals/pathogens is the immunological components such as S-IgA that associates with mucus [8]. Our studies used lavages from the murine URT as a source of mucus so that our findings could be correlated *in vivo* using murine models of colonization or hNF, collected and pooled from multiple donors, as a source of mucus from the natural niche of Spn. We took a broad view to interrogate Spn interaction with mucus by focusing on Spn surface factors. The capsule served as a positive control to validate our adherence assays [7]. Among the surface factors tested that had previously been implicated in mucus interactions, there was no role of a putative mucin binding protein MucBP or, in the absence of specific S-IgA using mucus from murine lavages, the Type 1 pilus [8, 13]. The lack of effect for MucBP could be accounted for by the differences in glycosylation between mucins of different species. We observed that Spn surface enzymes modify mucus to reduce bacterial adherence and that the major pneumococcal factors involved are the neuraminidases. Decreased mucus binding can be explained by a reduction

in the anionic and hydrophobic characteristics of mucus glycoconjugates through removal of terminal sialic acid residues or removal of residues to which Spn can bind. There were inconsistent effects with other exoglycosidases (StrH and BgaA), suggesting that the main mechanism affecting mucus interactions with Spn is through desialylation rather than further deglycosylation once terminal sialic acid is removed. EstA is reported to improve the efficiency of neuraminidases by deacetylation of sialic acid residues in a host species-dependent manner, but we were unable to confirm its role in the observed neuraminidase-dependent effects in our experimental approach [12].

It was unexpected that knocking out either *nanA* or *nanB* would increase mucus adherence and eliminate mucus desialylation and neuraminidase activity. This raised the question as to why Spn expresses two neuraminidases that function in in a non-redundant and non-additive manner. NanA has been established in the literature as a typical hydrolytic sialidase that cleaves α2,3-, α2,6- and α2,8-linked sialic acids to produce *N*-acetylneuraminic acid and as a Spn virulence factor [24, 29]. In contrast, a distinct role of NanB in the pathogenesis of Spn infection has been less clear. NanB has been reported to be an intramolecular trans-sialidase that acts to preferentially cleave α2,3-linked sialic acid substrates to release 2,7-anhydro-Neu5Ac and has been shown to function in the deglycosylation of host glycoconjugates [30, 31]. Both neuraminidases contain N-terminal signal sequences and are secreted enzymes. NanA, however, is cell wall-associated due to a typical C-terminal sortase-dependent anchoring motif, which is absent in NanB [24, 32, 33]. The lower neuraminidase activity observed in the Type 4 (TIGR4) strain could be due to a frame-shift prior to the LPxTG motif, eliminating cell-wall anchoring by sortase. In an outbred mouse model of infection, both NanA and NanB have been shown to be essential for colonization and infection of the upper and lower respiratory tract and for survival in the blood [29, 34]. However, in previous studies from our lab using infant rat models, NanA was shown to have no role in colonization [35].

Based on our findings, we propose the following model. Spn's upregulation of *nanA* expression requires sensing of host-derived sialic acid, but this enzyme is surface associated and free sialic acid is not readily available in its URT microniche [5]. To provide a source of sialic acid, it must be liberated from host glycoconjugates and this requires a secreted neuraminidase, NanB, encoded on a separate operon, *nan* II [25]. (Some strains also encode for NanC, which has many features in common with NanB). 2,7-anhydo-Neu5Ac released by NanB is internalized through an ABC transporter (SPD_1500-SPD_1502) also encoded by the *nan* II operon, as has been suggested for homologs in another bacterial species, *Ruminococcus gnavus* [36]. Once inside the cell, an oxidoreductase (SPD_1498) and a putative isomerase (SPD_1503), also encoded by *nan* II, covert 2,7-anhydo-Neu5Ac to Neu5Ac, again suggested by the role of a homologous oxidoreductase in *R. gnavus* [36]. The *nan* II operon, therefore, appears to function to liberate a source of sialic acid derived from host glycoconjugates to be taken up and sensed within the bacterial cell. Unlike other genes in the *nan* locus, the *nan* II operon is not regulated by *nanR*, which appears to act downstream on the other transcriptional units of the *nan* locus [25]. Internalized sialic acid produced by the combined activity of NanB and the other genes in the *nan* II operon might modify transcription control of *nanA* through its activator NanR. Since NanB does not influence the transcription of *nanR*, the regulator of *nanA*, this suggests that NanB-dependent transcriptional control of *nanA* is post-transcriptional with respect to *nanR* expression (Fig 4E). NanA is the main functional Spn neuraminidase as in its absence *nanB* is still expressed but there was no detectible evasion of mucus binding or desialylation of mucus. In isolates that lack *nanB* (4%), Spn may be able to scavenge free 2,7-anhydo-Neu5Ac released by other URT colonizers. If these strains still contain the rest of the *nan* II operon, they could internalize Neu5Ac to activate *nanR* and the transcription of *nanA*. So, even in the absence of *nanB*, the *nan* locus is hypothesized to be able to scavenge

free 2,7-anhydo-Neu5Ac to promote *nanA* expression. Thus, this report assigns a mechanistic role to 'the second neuraminidase' of Spn, NanB, and puts into perspective why prior studies were unable to account for its function apart from that of NanA.

Even though the expression of *nanA* and *nanB in vivo* has been established, previous *in vivo* studies examining the contribution of Spn's neuraminidases have yielded mixed results, [9, 29, 35, 37–39]. The lack of a colonization defect for the Δ*nanA*,*nanB*::janus mutant was surprising considering its robust mucus adherence phenotype. Interactions with mucus, however, are likely to be complex. The first encounter of Spn with its host is with 'loose' luminal mucus [2]. Our finding that the neuraminidase-deficient mutant shows an early competitive advantage suggests that attaching to this material might aid in retention of the bacterial inoculum. However, during the period of stable colonization (>24hrs p.i.), when Spn has migrated to the glycocalyx, we confirmed that the presence of Spn increases URT secretions, a result that recapitulates the correlation between rhinitis symptoms and Spn carriage in young children [40, 41]. Adults, who have a lower density of colonizing Spn, have less pronounced secretions during episodes of carriage. In infant mice, increased secretions in URT lavages and along the mucosa (seen as a thicker glycocalyx layer) depended on the expression of its neuraminidases with deletion of either eliminating this effect as NanB controls expression of *nanA*. The neuraminidase-mediated increase in URT secretions was independent any effects of the neuraminidases on levels of Spn colonization. It appears, therefore, that the neuraminidase-mediated stimulation of mucus which could otherwise sweep away colonizing pneumococci is balanced by modulation of mucus by neuraminidases to limit bacterial binding and removal by mucociliary clearance. It remains unclear how bacterial neuraminidase acts on the mucosa to increase mucus production and flow. Since neuraminidases mediate both an effect on mucus production and evasion it is difficult to test conditions that could distinguish these effects. Interestingly, influenza virus, which contains a neuraminidase, induces copious mucus production in the URT [5, 40]. In the setting of influenza A co-infection, the neuraminidase-deficient Spn mutant has been shown to exhibit decreased levels of colonization compared to controls without influenza A co-infection [5, 42]. We postulate that this is due to the inability of the mutant to evade mucus binding when mucus secretions are abundant, although we cannot exclude other effects of neuraminidases such as providing a source of sialic acid for nutritional purposes. [6, 31]. Additionally, in previous TnSeq screens of Spn genes affecting mouse colonization, we and others observed selection against *nanB* (P = 0.01) and *nanA* (P = 0.029) mutants [43, 44]. This could be explained by the decreased mucus evasion of these mutants under conditions where mucus production is stimulated by co-colonizing strains still expressing neuraminidase. A further consideration is that Spn may need to fine tune mucus production and binding to allow it to be shed from its host in URT secretions in the process of transmission to a new host.

In summary, by studying Spn-mucus interactions we found that Spn expresses at least two neuraminidases because one is required for the expression of the other–an unusual scenario where an organism's ability to target a host substrate depends on an enzyme with related function. In particular, our findings show that the Spn neuraminidases act together to desialylate mucus to limit bacterial binding. We postulate that during colonization neuraminidase-dependent reduction in mucus binding facilitates evasion of mucociliary clearance which is needed to counter neuraminidase-mediated stimulation of mucus secretions.

## Materials and methods

### Ethics statement

All animal experiments followed the guidelines summarized by the National Science Foundation Animal Welfare Act (AWA) and the Public Health Service Policy on the Humane Care

and Use of Laboratory Animals. The Institutional Animal Care and Use Committee (IACUC) at New York University School of Medicine oversees the welfare, well-being, proper care and use of all animals, and they have approved the protocol used in this study (IA16-00538).

## Mice

C57BL/6J mice were purchased from The Jackson Laboratory (Bar Harbor, ME), and were bred and housed in a conventional animal facility. Animals were fed ad lib the PicoLab Rodent Diet 20, which is a 20% protein diet formulation and were given acidified water for consumption. Additionally, the animals were kept on a light-cycle of 12 hours on, 12 hours off with a temperature in the animal facility of 70˚F (±2˚F). Throughout the experiments, mice remained healthy and did not lose weight compared to uninfected controls.

## Chemicals and reagents

Bovine serum albumin (BSA; A9430), sodium periodate (Cat. No. 71859), O-phenylenedia-mine dihydrochloride (Cat. No. P9187), cholera filtrate lyophilized powder (Cat. No. C8772), mutanolysin (Cat. No. M9901), lysozyme (L6876), Tween 20 (Cat. No. P9416), Trypsin (Cat. No. 85450C) and Alcian blue solution (Cat. No. B8438) were obtained from Millipore Sigma (Darmstadt, Germany). Rabbit anti-pneumococcus Type 4 serum (Cat. No. 16747) and Type 23F serum (Cat. No. 16913) were obtained from Statens Serum Institut (Copenhagen, Denmark). Triton X-100 (9002-93-1) was obtained from Amresco (Solon, OH, USA). HRP-coupled streptavidin (Cat. No. 21130), 4% paraformaldehyde solution (Cat. No. AAJ19943K2), NA-STAR kit to assay neuraminidase activity (Cat. No. 4374422), High-Capacity cDNA Reverse Transcription Kit (Cat. Not. 4368814), Power SYBR Green PCR Master Mix (Cat. No. 4374967), 2X Phusion Master Mix (Cat. No. F531L), Streptavidin- Alexa Fluor 488 (Thermo Fisher (Invitrogen), Cat #32354), ProLong Gold mounting reagent (Thermo Fisher, Cat #P10144) and Dulbecco′s Modified-Eagle′s Medium (DMEM, Cat. No. 11995–065) were obtained from Thermo Fisher Scientific (Waltham, MA, USA). FITC-conjugated mouse anti-goat antibody (Cat. No. sc-2356) was purchased from Santa Cruz (Dallas. TX, USA). Bradford Reagent (Cat. No. 500–0006) was obtained from Bio-Rad (Hercules, CA, USA). Biotinylated Maackia Amurensis Lectin II (MAL II, B-1265), Biotinylated Sambucus Nigra Lectin (SNL, EBL, B-1305), Biotinylated Maackia Amurensis Lectin I (MAL I, B-1315), and Carbo-Free blocking agent (Cat. No. SP-5040) were purchased from Vector Laboratories (Burlingame, CA, USA). Biotin-labelled Mucin 5AB-1 (45M1) was purchased from NeoMarkers Inc. (Portsmouth, NH, USA). RNAprotect cell reagent (Cat. No. 76526) and RNeasy (Cat No./ID: 74106) were obtained from Qiagen (Hilden, Germany). MasterPure DNA purification kit (Cat. No. MCD85201) was obtained from Epicentre (Middleton, WI, USA). GoTaq Green Master Mix (Cat. No. M7123) was obtained from Promega (Madison, WI, US).

## Bacterial culture

The Type 4 TIGR4 (T4) and Type 23F (T23F) *Streptococcus pneumoniae* (Spn) strains that were used in this study are listed in S1 Table. Spn isolate P2406 (streptomycin resistant [str[r]]) is a capsule type 4 (T4) derivative of TIGR4 and strain P2499 is a Str[r] derivative of P1121, a capsule type 23F (T23F) strain isolated from the nasopharynx in a human experimental carriage study [45, 46]. Spn were grown on tryptic soy (TS; Becton Dickinson) agar plates supplemented with 100 μl of catalase (30,000 U/ml; Worthington Biomedical) and appropriate antibiotics (200 μg/ml streptomycin, str; 125 or 250 μg/ml kanamycin, kan; 2 μg/ml chloramphenicol, 1 μg/ml erythromycin, or 200 μg/ml spectinomycin), overnight at 37˚C and 5% $CO_2$. The number of Spn (CFU/mL) was confirmed for each assay described below by plating serial

dilutions on TS agar (supplemented with the appropriate antibiotic). Broth-grown Spn were obtained by static culture in TS broth at 37°C to an $OD_{620}$ of 1.0 or 0.6 for *in vivo* and *in vitro* experiments, respectively, unless otherwise specified. Spn were centrifuged at 10,000xg for 1 min, washed, and diluted in sterile PBS to the desired concentration.

## Bacterial strain construction

The primers used to construct all of the bacterial strains are listed in S2 Table.

The in-frame and unmarked deletion pneumococcal strains, deficient for the genes *nanA*, *nanB*, *mucBP*, *estA*, and *strH*, were constructed in a two-step process using the Janus cassette [47]. First, genomic DNA from strain P2408, containing the Janus cassette, was isolated using the MasterPure DNA purification kit (MCD85201 Epicentre, Middleton, WI, USA). For constructing a *nanB*-deletion mutant, flanking regions 1 kb upstream and downstream of the Spn *nanB* gene were added to the Janus cassette via isothermal assembly (using primers 83F and 84R, 85F and 86R, and 87F and 88R) [48]. The PCR product was then transformed into the str^R T4 parent strain Spn P2406 and the transformants were selected on TS-kan (250 μg/ml) plates. The presence of the Janus cassette in the *nanB* gene in P2406 was confirmed by PCR (GO-Taq polymerase, Promega M7123, and primers 83F and 88R). A second PCR amplicon was generated by amplifying and joining the upstream and downstream regions around *nanB* from Spn P2406 (using primers 1F and 2R and primers 3F and 4R). Spn P2613, the intermediate strain (*nanB*::janus; kan^R, str^S), was transformed with this PCR product to generate an in-frame, unmarked *nanB* deletion strain. Transformants were selected on TS-str to generate strain P2619 (Δ*nanB*; str^R, kan^S) and was confirmed by PCR (GO-Taq polymerase using primers 83F and 88R). The in-frame knockout strain P2619 has a scar containing the first and last 5-amino-acid coding sequences of the *nanB* gene. The *nanB* corrected strain P2623 (Δ*nanB*::*nanB*; str^R, kan^S), was constructed by transforming P2613 (*nanB*::janus*)* with a complete *nanB* PCR product generated using primers 1F and 4R on strain P2406; the transformants were selected on TS-str. None of the constructed strains showed a growth defect when cultured in TS broth. This same procedure was used to a create in-frame, unmarked deletion strains of *nanB* and the chromosomally corrected mutant in Spn T23F P1121. These methods were also used to create *nanA*, *nanB* deficient strains and their respected chromosomally corrected strains in the previously-described Spn Type 23F::*pilus-1* P2588 strain [8, 49]. The Spn strains that were generated in this study are listed in S1 Table.

To create a Spn strain with an enzymatically-inactive NanB, a PCR product was generated (using the primer pairs consisting of 79F and 81R and 80F and 82R) to amplify *nanB* from Spn P2499 (a strR derivative of P1121). The primers were constructed to replace an asparagine at amino acid 270 with an alanine. The PCR fragment was transformed into Spn P2637 (Δ*nanB*::janus; str^S, kan^R), transformants were selected on TS-str (200 μg/ml) and confirmed by PCR (GO-Taq polymerase using primers 1F and 4R). The mutation was validated by Sanger sequencing.

## Human and mouse nasal fluid binding assays

Pooled nasal secretion samples from six adult volunteers were purchased from LeeBio (Maryland Heights, MO, USA, 991-13-S). Human nasal fluid samples were sonicated on ice (one second at amplitude of 8 um; Fisher Scientific Model 705 Sonic Dismembrator) to create homogenous samples, pooled and stored at -20°C. The amount of protein per sample was determined through a protein concentration assay (Bio-Rad Protein Assay, Cat. # 500–0006) per manufacturer's guidelines. Briefly, 10uL of samples were mixed with 200 μL of diluted dye reagent (1:5) and incubated at RT for 10 min. Absorbance was measured at 595 nm using a

SpectraMax M3 plate reader (Molecular Devices). The total protein for human nasal fluid samples was equalized to approximately 10μg/100μL. For example, if the total protein per sample was 0.143mg/mL, for 4 wells, we would need 40μg of protein and 400μL of fluid (100μL needed per well). We would use 279.7 μL of human nasal fluid diluted in 120.3μL of PBS (pH 7.4). Murine nasal lavages (mNL) were collected from uninfected, adult C57BL/6J mice through post-mortem retro-tracheal lavages with 400uL sterile PBS (pH 7.4), pooled (minimum 40 mice) and stored at -20˚C.

Adherence of different pneumococcal strains to hNF or mNL's was assessed in a solid phase-binding assay as previously described [8]. Briefly, 100μL /well of hNF (10μg of total protein diluted in PBS) or mNL (undiluted) were immobilized on a 96-well flat-bottom polystyrene plate (Sarstedt REF:82.1581.001) by centrifugation (250 x g for 3min, RT) and incubated overnight at 37˚C in 5% $CO_2$. Wells were then washed 3 times with 100μL of DMEM and blocked with 0.1% BSA-DMEM for 2h at RT. The wells were then washed 3 times with 100μL of DMEM and Spn (2 x $10^4$ CFU/mL in 100μL of DMEM, grown to mid-log phase ($OD_{620}$ = 0.6) in TS broth and then diluted to the desired concentration) were added to each well. The plate was then centrifuged (250 x g, 3min, RT) and the plate was incubated at 30˚C and 5% $CO_2$ for 2h to allow the Spn to bind. Then, to remove unbound bacteria, the wells were gently washed 19 times with 100μL of DMEM. Adherent bacteria were collected by adding 100 μl of 0.001% Triton X100-PBS and incubating for 15 min at RT followed by vigorous mixing. To quantify the adherent bacteria, samples were plated in triplicate on TS agar plates with select antibiotics and incubated overnight at 37˚C with 5% $CO_2$. Each experimental condition was assayed in triplicate for each experiment.

In some assays, the mNL's were modified after immobilization. To assess the role of carbohydrate oxidation, immobilized mNL was pretreated for 30 min with 100μL of 100mM of sodium periodate ($NaIO_4$, Sigma, Cat. No. 71859) in 50mM sodium acetate buffer (pH 4.5) and then blocked with 0.1% BSA-DMEM for 2h at RT. The high concentration of $NaIO_4$ (100mM) provides non-specific oxidation of sugar moieties [50]. To assess the role of protein cleavage, immobilized mNLs were treated with 50μg/ml trypsin (Sigma, Cat. No. 85450C) in 100uL PBS and then blocked with 0.1% BSA-DMEM for 2h at RT.

For neuraminidase complementation studies, immobilized hNF were treated with 200μL cholera filtrate (a source of neuraminidase, Sigma Aldrich, C8772), resuspended in calcium saline solution at 100 U/mL and incubated overnight at 37˚C and 5% $CO_2$. The wells were then washed 3 times with 100μL DMEM and blocked with 0.1% BSA-DMEM for 2h at RT.

## Sialic acid quantification ELISA

An ELISA was used to measure the removal of hNF sialic acid by Spn. Wells of a microtiter plate (96-well, Immulon 4HBX plate, Thermo Fisher Scientific) were coated with 100μL hNF diluted in PBS (pH 7.4) and incubated overnight at 37˚C and 5% $CO_2$. The wells were washed 3 times with 100μL DMEM, and blocked with 200μL of Carbo-Free Blocking Solution (Vector Laboratories, SP-5040) for 2h at RT. The wells were washed 3 times with 100μL of DMEM. Spn [100μL of 5 x $10^6$ CFU/mL Spn grown to mid-log phase ($OD_{620}$ = 0.6) and diluted to the desired concentration in DMEM] were added to the wells and the plate was incubated for 4h at 30˚C and 5% $CO_2$. The wells were then gently washed 19 times with 100μL of washing buffer TPBS (PBS + 0.01% Tween 20). Remaining -2,3-linked sialic acid was detected by adding 100μL biotin-linked Mal I (2.5μg/mL, Vector Laboratories, B-1265) and incubating for 1h at RT. Wells were washed 3 times with 100μL TPBS and incubated with streptavidin-horseradish peroxidase (HRP)-conjugate (Pierce #21130) diluted 1:5,000 in PBS for 1h at RT. Wells were washed three times with 100μL TPBS and O-phenylenediamine dihydrochloride (Sigma,

P9187) was used as an HRP substrate, according to manufacturer's directions. The color reaction was measured at an absorbance of 492 nm using a SpectraMax M3 plate reader (Molecular Devices). The absorbance of the PBS control wells was averaged and the value was subtracted from each measured experimental value.

## Microscopy

For microscopic visualization of mNLs and hNF, nasal fluid samples were processed and assessed as previously described [8]. Briefly, 1μL of 5 x $10^4$ CFU/mL Spn were incubated with 10 μl of undiluted hNF or mNL for 2 h at 37˚C and 5% $CO_2$. The total volume of sample (11μL) was then placed onto glass slides and heat fixed by flame. To visualize mucus, samples were incubated with 3% acetic acid for 5 min followed by Alcian blue (Sigma, B8438) (in 3% acetic acid, pH 2.5) for 30 min. After washing the samples in sterile water for 10 min, slides were blocked in 10% fetal bovine serum (FBS; Peak Serum PS-FB1) in PBS at 4˚C overnight. Bacteria were stained with rabbit anti-capsule (Type 4 (Statens Serum Institut, 16747) or Type 23F serum (Statens Serum Institut, 16913) antibody (1:200 in 0.5% FBS-PBS) and secondary goat anti-rabbit IgG-FITC (1:100 in 0.5% FBS-PBS) (Santa Cruz, Cat. No. sc-2356). Spn were visualized on an Axiovert 40 CFL microscope with an Axiocam IC digital camera (Zeiss). Images were analyzed with ZEN 2012 software and, for brightness and contrast, processed with ImageJ 1.52a software.

## Neuraminidase activity assay

To determine the neuraminidase activity for the Spn strains, we used the NA-STAR kit from Thermo-Fischer (4374422) according to the manufacturer's protocol. In summary, bacteria were grown to an $OD_{620}$ = 1.0, centrifuged at 10,000 x *g* for 1min and resuspended in 250μL of PBS to a density of $10^7$ CFU/mL. Bacterial suspensions were sonicated on ice (15s on, 45s off for 8 min total; Amplitude 8 μm; Fisher Scientific Model 705 Sonic Dismembrator). After sonication, the samples were centrifuged at 10,000 x *g* for 1 min, and the supernatant was kept on ice until neuraminidase activity was assayed. Using the NA-STAR kit plates, 25μL of NA-Star assay buffer was added to both bacterial and control wells. Next, 50μL of PBS was added to the control wells and 50μL of sample was added to the experimental bacterial wells. The plate was incubated for 20 min at 37˚C and 5% $CO_2$. Next, 10μL of diluted NA-STAR substrate in NA-STAR buffer was added to each well and the plate was incubated for 30 min at RT. 60μL of NA-STAR accelerator was added to each well and the chemiluminescent signal (Luminescence program, check LM1 all) was read by a SpectraMax M3 reader (Molecular Devices). To determine the neuraminidase activity from the experimental wells, the "signal" (luminescence value for each sample) was divided by the "noise" (luminescence value of PBS control wells).

To assess how the presence of sialic acid in mNL's might impact neuraminidase activity of Spn strains, we performed the following experiment. Briefly, starting at an $OD_{620}$ = 0.2, Spn were grown in TS broth for 45 min; 3 mL of culture was then centrifuged at 10,000x*g* for 1 min, and the pellet was resuspended in 3mL DMEM. The bacteria were diluted 1:50 into 3mL DMEM alone, or into 3mL DMEM with 5% of mNL. The samples were incubated for 3h at 37˚C with 5% $CO_2$. The samples were then centrifuged at 10,000 x g for 1 min, resuspended in 250μL PBS, and neuraminidase activity assayed as described above.

## Quantitative RT-PCR

Spn were grown in TS broth to an $OD_{620}$ = 1.0. Samples were mixed with an equal volume of RNA protect (Qiagen, 76526) and incubated for 5 min at RT followed by centrifugation for 1 min at 10,000 x g; the bacterial pellet was stored at -80˚C. To extract RNA, Spn pellets were thawed and treated with 27μL mutanolysin (200 mg/mL, Sigma M9901), 18μL Proteinase K

(20 mg/ml, Denville Scientific CB3210-5), 27μL of lysozyme (100 mg/mL, Sigma L6876), and 128μL of TE buffer (10mM TrisCl, 1mM EDTA, pH 8.0, nuclease-free water) for 20 min at RT. RNA extraction (RNeasy, Qiagen) and subsequent cDNA generation (High Capacity cDNA Reverse Transcriptase Kit, Applied Biosystems, Thermo Fisher Scientific), were performed according to manufacturer's instructions. PCR with *gapdh* was performed on purified RNA to check for DNA contamination. qRT-PCR reaction samples contained ~10ng cDNA and 0.5 μM primers in Power SYBR Green PCR Master Mix (Applied Biosystems, Thermo Fisher Scientific) and samples were tested in duplicate. qRT-PCR reactions were run in a 384 well plate (Bio-Rad) using CFX384 Real-Time System (Bio-Rad). Expression of *16S* and *gapdh* were normalization controls and fold-change in gene expression was quantified according to the ΔΔCt method [51]. The primer sequences used in this assay are indicated in S3 Table.

## Mouse infections

Spn strains were grown in TS broth to an $OD_{620} = 1$, washed, and diluted to the desired density in sterile PBS. Six-week old adult mice were infected intranasally without anesthesia, with 10μL containing ~1–2 x $10^5$ CFU of either Spn strain P2588 (Type 23F::*pilus-1*) or P2636, the isogenic neuraminidase-deficient strain (Type 23F, Δpilus-1, Δ*nanA*,*nanB*::janus) alone, or with an inoculum containing a 1:1 ratio of P2588 and P2636. Four-day old mice were infected intranasally without anesthesia, with 3μL containing 1-2x $10^3$ CFU of either Spn strain P2588 or P2636. At 4h, 24h, and 5 days post-pneumococcal challenge, mice were euthanized with $CO_2$ followed by cardiac puncture. For quantification of URT colonization density, the trachea was cannulated and lavaged with 200μL sterile PBS, and fluid was collected from the nares. The nasal lavage samples were plated in serial dilutions on TS-str or TS-kan (250μg/mL) plates and incubated overnight at 37 ºC with 5% $CO_2$.

## Immunoblot

An immunoblot using mNL's obtained from Spn-colonized mice provided a quantitative assessment of α2,3-linked sialic acid or muc5ac levels. Four-day old pups were infected with P2588 (Type 23F::*pilus-1*), P2636 (Δ*nanA*, *nanB*::janus), P2634 (Δ*nanA*), P2637(*nanB*::janus), P2642 (NanB$_{D270A}$) as described above, or mock-infected with PBS as a control. URT lavages were obtained 5d p.i., and stored at -20˚C until use. mNL's were diluted 1:5 in PBS and 100μL was applied to a 0.2-μm nitrocellulose membrane (Amersham Protran 0.2μm GE10600094) with vacuum using a Minifold II Slot- Blot apparatus (Schleicher & Schuell). Air dried membranes were incubated in 2% BSA-Tris-buffered saline (TBS) at 4˚C overnight with shaking. The membrane was incubated for 1h at RT in 2μg/mL Maackia Amurensis Lectin I biotinylated (Vector Laboratories #B-1315) or 1:500 of muc5ac monoclonal antibody (NeoMarkers #45M1) diluted in TBS. The blot was washed 6 times (5 min each wash) with 0.1% Tween-20-TBS, and incubated for 1h at RT with streptavidin-horseradish peroxidase (HRP)-conjugate (Pierce #21130) diluted 1:100,000 in TBS. The membrane was washed once for 1.5 h, followed by 5 times (15 min each wash) with 0.1% Tween-20-TBS and was developed, according to manufacturer's directions, with the Super Signal West Femto substrate (Thermo Scientific Cat. No. 34095). The chemiluminescent signal was visualized using the iBright CL1000 imaging system (Thermo Fisher Scientific) and the relative intensities of the bands on the membrane were quantified through assessing integrated pixel density.

## Histopathology

Pups infected as described above were euthanized as per standard protocol at day 9 (5 days p. i.). The skin of heads were gently removed to preserve nasal structures. Heads were then decapitated and submerged in PBS at 4˚C for a brief wash, followed by fixing in 4% paraformaldehyde for 72h at 4˚C without shaking. Heads were then washed in PBS at 4˚C with gentle

swirling x 20 min, followed by decalcification by fully submerging heads into 0.12M EDTA solution at 4°C with gentle shaking for 7 days. Intact heads were then processed through graded ethanols to xylene and infiltrated with paraffin in a Leica Peloris automated tissue processor. Five um paraffin-embedded sections were stained either with PAS and alcian blue (Freida L Carson, Histotechnology 2nd Ed., 1997) or with lectin and antibody probes on a Leica BondRX automated stainer, according to the manufacturer's instructions. In brief, sections were incubated for 2h with Sambucus Nigra Lectin (SNL-EBL) conjugated to Cy5 (1:50 dilution, Vector Labs, cat # CL-1303) followed by 1h with Spn typing sera (1:2000 dilution, SSI diagnostica, cat # 16913) and then 1h with goat-anti rabbit IgG conjugated to Alexa Fluor 594 (1:100, ThermoFisher A21207). Slides were scanned on an Akoya Polaris Vectra imaging system. The multispectral images were unmixed and autofluorescence signal removed with the Akoya InForm software prior to exporting as tif files.

Human nasal fluid samples were processed and assessed as previously described [8]. Briefly, 1μL of 5 x $10^4$ CFU/mL Spn were incubated with 10 μl of undiluted hNF for 2 h at 37°C and 5% $CO_2$. The total volume of sample (11μL) was then placed onto glass slides and heat fixed. Initially, antigen retrieval was performed for 10min in buffer pH 6.0 at 100°C. Staining was done using an automatic stainer Leica BOND RX. Primary antibody incubations were for 60 min at RT with Muc5AC monoclonal antibody conjugated to biotin (NeoMarkers #45M1) at 1:100 or with Spn typing sera (SSI diagnostica, cat # 16913) at 1:2000. Secondary antibody incubation was 60 min at RT with Streptavidin- Alexa Fluor 488 (Thermo Fisher (Invitrogen), Cat #32354) at 1:100 or Anti Rabbit IgG, Alexa Fluor 494 (Thermo Fisher, Cat #A21207) at 1:100. Slides were counterstained with DAPI (5', RT) and mounted using ProLong Gold mounting reagent (Thermo Fisher, Cat #P10144). Spn were visualized by microscopy on a LM- Zeiss AxioObserver at 63x and analyzed through Zen light software.

## Statistical analysis

GraphPad Prism (version 7.01, San Diego, CA) was used for statistical analysis. A one-way ANOVA, using Dunnett's multiple comparison test or a Kruskal-Wallis test with Dunn's multiple comparison test, was used to measure differences in means of 2 or more groups. An unpaired T-test was performed to compared two data sets that passed normality tests, while a Mann-Whitney test was used if the data did not pass normality tests. A sample t and Wilcoxon test was employed to compare data to a theoretical mean of 1, where noted.

## Supporting information

**S1 Fig. Representative Immunoblot for detection of α2,3 linked sialic acid.** To quantify sialic acid and mucus containing secretions in the URT of mice, retro-tracheal lavages were obtained from infant mice at day 5 p.i.. After collection, using a slot-blot manifold apparatus an immunoblot was performed to determine the amount of α-2,3 linked sialic acid present in different infections. There are 3 infection types, mock-infected, Type 23F::*pilus-1* and *ΔnanA*, *nanB*::janus. Each infection type is shown for two animals and each shown in triplicate with individual animals denoted by the type of infection and replicate. Immunoblots were used for quantification by densitometric analysis.
(DOCX)

**S1 Table. Bacterial Strains used for the study.** Bacterial strains used in this study are outlined in S1 Table. Information includes the serotype, genotype, antibiotic resistance and the source or reference for the strain.
(DOCX)

**S2 Table. Primers used for bacterial strain construction.** Bacterial primers used in this study for strain construction are outlined in S2 Table. Information includes the gene target, primer name and Sequence (5'→3') for the primer.
(DOCX)

**S3 Table. Primers used for quantitative RT-PCR in this study.** Bacterial primers used in the study for quantitative RT-PCR are outlined in S3 Table. Information includes the gene target, primer name and Sequence (5'→3') for the primer.
(DOCX)

## Acknowledgments

We would like to thank the NYU Experimental Pathology Research Laboratory for assisting with the histopathology studies. The core is partially supported by the Cancer Center Support Grant P30CA016087 at NYU Langone's Laura and Isaac Perlmutter Cancer Center and the Vectra Imaging system was awarded as Shared Instrumentation Grant S10 OD021747. We would also like to thank NYU Langone's Microscopy Laboratory for assisting with the microscopy studies. The core is partially supported by the Cancer Center Support Grant P30CA016087 at the Laura and Isaac Perlmutter Cancer Center.

## Author Contributions

**Conceptualization:** Alexandria J. Hammond, Ulrike Binsker, Surya D. Aggarwal, Mila Brum Ortigoza, Jeffrey N. Weiser.

**Data curation:** Alexandria J. Hammond.

**Formal analysis:** Alexandria J. Hammond, Surya D. Aggarwal, Mila Brum Ortigoza, Cynthia Loomis, Jeffrey N. Weiser.

**Funding acquisition:** Jeffrey N. Weiser.

**Investigation:** Alexandria J. Hammond, Ulrike Binsker, Surya D. Aggarwal, Mila Brum Ortigoza, Jeffrey N. Weiser.

**Methodology:** Alexandria J. Hammond, Ulrike Binsker, Surya D. Aggarwal, Mila Brum Ortigoza, Cynthia Loomis, Jeffrey N. Weiser.

**Project administration:** Jeffrey N. Weiser.

**Resources:** Jeffrey N. Weiser.

**Supervision:** Jeffrey N. Weiser.

**Validation:** Alexandria J. Hammond.

**Visualization:** Alexandria J. Hammond, Mila Brum Ortigoza, Cynthia Loomis.

**Writing – original draft:** Alexandria J. Hammond, Jeffrey N. Weiser.

**Writing – review & editing:** Alexandria J. Hammond, Ulrike Binsker, Surya D. Aggarwal, Mila Brum Ortigoza, Jeffrey N. Weiser.

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
