## [Decision Letter · Decision Letter 0]

27 Dec 2020

Dear Dr. Weiser,

Thank you very much for submitting your manuscript "Neuraminidase B controls neuraminidase A-dependent mucus production and evasion" for consideration at PLOS Pathogens. As with all papers reviewed by the journal, your manuscript was reviewed by members of the editorial board and by several independent reviewers. In light of the reviews (below this email), we would like to invite the resubmission of a significantly-revised version that takes into account the reviewers' comments.

The reviewers were generally very positive towards the manuscript. However meaningful concerns regarding the methodology used in mucus collection and the integrity of the mucus scaffold during experimentation require attention. It is important that the mucus model be validated prior to resubmission. Additionally, better explanation of the statistical tests chosen and clarity in the legends is requested.

We cannot make any decision about publication until we have seen the revised manuscript and your response to the reviewers' comments. Your revised manuscript is also likely to be sent to reviewers for further evaluation.

Sincerely,

Carlos Javier Orihuela, PhD

Associate Editor

PLOS Pathogens

Michael Wessels

Section Editor

PLOS Pathogens

Kasturi Haldar

Editor-in-Chief

PLOS Pathogens

orcid.org/0000-0001-5065-158X

Michael Malim

Editor-in-Chief

PLOS Pathogens

orcid.org/0000-0002-7699-2064

The reviewers were generally very positive towards the manuscript. However meaningful concerns regarding the methodology used in mucus collection and the integrity of the mucus scaffold during experimentation require attention. It is important that the mucus model be validated prior to resubmission. Additionally, better explanation of the statistical tests chosen and clarity in the legends is requested.

Reviewer's Responses to Questions

**Part I - Summary**

Reviewer #1: This is an interesting paper that explores a relevant topic: how bacteria evades mucociliary clearance as a defense mechanism. The experiments conducted and discussed are relevant and significant. Overall, the paper is strong, with one major weakness in the characterization of the mucus samples, as outlined below.

Reviewer #2: Comments for the authors of the PLoS Pathogens manuscript number PPATHOGENS-D-20-02578:

The authors of the PLoS Pathogens manuscript “Neuraminidase B controls neuraminidase A-dependent mucus production and evasion”, presents some very interesting findings related to host:pathogen interactions with Streptococcus pneumoniae. Specifically, this group has been using their recently-developed solid-phase adherence assay to evaluate Spn adherence as they study bacterial interactions with the host in the presence of mucus. This manuscript shows that the neuraminidase genes NanA and NanB both contribute to mucus evasion through removal of terminal sialic acids and that, interestingly, NanB controls NanA expression. This process works through NanB releasing sialic acid residues that then activate NanA activity. Ultimately, NanA performs the bulk of the sialic acid modifications, while NanB is required to get the process started. The group did an excellent job of identifying the specific contributions of NanA and NanB to this process, including demonstrating that this was not controlled by NanR. This well-designed, well-controlled study presents some interesting results that are described with appreciation for how these findings advance the field. There was only one minor typographical error that needs to be corrected, and that is listed below.

Reviewer #3: This is an interesting and generally well written manuscript that addresses the long-standing question of what the biological function of NanB might be, given that all previously known contributions of pneumococcal neuraminidase are attributed to NanA. The experiments and data are for the main part logical and support the conclusions of the Authors. These are complex models and so it is not surprising that there is a lot of variability in some of the data. However, a range of statistical tests are applied, and the logic is not explained. This leaves one wondering if there is a reason or if this was just to achieve statistical significance. There are other relatively minor concerns regarding the clarity of legends and strain names, the interpretation of data and the conclusions drawn. A more in-depth discussion of what might occur in nanB negative isolates (especially those that also lack nanC) would be interesting. Those strains still appear to maintain the rest of the nanB operon – making this Reviewer wonder if they scavenge 2,7-anhydroneuraminic acid released by other bacteria.

**Part II – Major Issues: Key Experiments Required for Acceptance**

Reviewer #1: 1. Mucin methods are very non-specific. The nasal washes, while appropriate to test the question, as Spn colonized the nasal cavity, will capture many things in the mucus besides mucin. The washes are not well described. What is the concentration of the mucus in the washes? How many cells are captured in the wash? Are they removed?

2. Sonication will disrupt mucin polymers, causing the sample to not be native as characterized in the manuscript. Disruption of polymers will change the availability of the side chains, including the sialyated groups, for the Spn to act on enzymatically. Validation data demonstrating that sonication is not influencing the results is necessary.

3. In Figure 1E, there is no way to identify the faint blue dots as mucus particles. The panel needs a scale bar is needed for this panel. IHC for Muc5ac, as done in the later figures, is necessary here.

Reviewer #2: None required.

Reviewer #3: None

**Part III – Minor Issues: Editorial and Data Presentation Modifications**

Reviewer #1: 1. Table 1 is confusing, especially the second part with all the foot notes. Why were some mutations only tested in one strain? Some of the data suggests that the strain is important. This might be better represented as a bar graph.

2. In line 159, do you intend to refer to Figure 2C?

3. In the mouse, the protein Muc5ac is noted with lowercase letters.

4. Line 335 needs a citation, following “rather than native mucus”. Many groups are studying native mucus.

5. A statement regarding food and water, temperature, and light cycle of the animal facility would aid the methods section.

Reviewer #2: General Comments:

1. On line 331, the sentence that reads: “…often found in embedded in the …” needs editing.

Reviewer #3: Please provide in the manuscript the logic for the statistical tests selected for different experiments.

6B is misleading – the different time points are done with different animal models and should be plotted separately or at least the distinction made clear.

Conclusions and interpretation:

A couple of observations would benefit from more discussion. What do the Authors propose to occur in isolates that naturally lack nanB (and nanC?). Furthermore, is the Authors explanation for the decreased colonization only in co-infection. I am struggling to generate a reasonable hypothesis to explain that.

Figure 5B. While the rest of the figure is convincing. It is not clear why the nanA complement does not complement the and no removal of sialic acid is observed. This should be discussed, but does not appear to be mentioned in the text at all.

Figure 6A. do the authors know that with these specific strains the mutant has a temporary advantage in binding to mucin in the animal model (line 292)? No longer time points were reported.

Figure 7A. The Authors mention that similar numbers of bacteria are seen in the top panel. The figure legend doesn’t mention this, but I assume it is the orange staining and the sialic acid green. If that is the case, can conclusions really be drawn from these images there are some significant differences observed. The CFU counts are informative but based on a single image and no quantification I don’t believe these are.

Figure 7B and C: Some of these differences are very small – an image of the blot and the boundaries drawn when assessing intensity would be useful for interpretation. Also, the methods say intensity of bands were measured, but were there bands? The rest of the methods read like a dot-blot

While the Authors are right, they saw no contribution of MucBP to nasal fluid (Line 344), I believe this experiment was only done with mouse nasal fluid. If that is the case it seems important to acknowledge that the lack of a phenotype could be due to differences in glycosylation between the mucins of different species.

Writing:

The manuscript is for the most part well written. An exception to this are parts of the methods, the strain names and the figure legends. The majority of the comments relative the legends are included below. But as an example

Figure 4. It is not clear from the legend and figure what data in panels C, D and E are plotted relative to. The panels all say wild-type, but the legend says parental strain for C, nanB for D and nanR for E. Is the fold-change relative to mutants in these genes? This doesn’t seem to match the data. Please clarify and make any necessary changes.

The strain names used throughout the paper lack consistency and clarity - the strain names in the strain table and methods don't match the text and mutant names in figures and some of the text give no indication of strain background. This makes it harder to follow. Is it possible to modify this and make the figures and text clearer?

Minor issues

Line 30: Should it read neuraminidase A not just neuraminidase?

What is the 23F strain? What site was it isolated from etc.

Line 102: While the point of this sentence is clear after reading it a couple of times it doesn’t seem to make sense as written. I believe a revision could make it clearer for the reader.

Figures: some of the lines indicating which conditions were compared by statistical analysis do not align with the bars – for example 1A the NS line actually falls over the 30 second time point bar on the graph.

Lines 144-5: I assume the authors looked at unbound bacteria under the microscope? Figure 1E does not show bound bacteria are in longer chains than unbound.

Line 180: Is NanA present in all strains of nanA?

Line 197: I assume the Type 23F strain naturally lacks the pilus locus? Perhaps that could be more clearly communicated.

Line 248: Should it be functional in place of function?

Line 487: Is this Table S1?

Line 626: Doesn’t make sense as written.

Line 656: Is this strain a pilus deletion or insertion?

Line 893: followed by not following?

Figure 5A: It might just be me, but it took a long time for me to get my head around this; although, the concept is simple. Is there another way of reporting the data or describing it in the text that might make it easier for the reader to understand the Authors point?

Legend figure 7: The figure legend doesn’t mention this, but I assume the orange staining are the bacteria and the sialic acid green. The legend reads as if immunoblots are shown. Furthermore, it says 2,6 linked sialic acid is shown in panel B, whereas the axis says 2,3. Which statistics are applied in each case are also not reported.

Figure 7. The Authors mention that similar numbers of bacteria are seen in the top panel.

PLOS authors have the option to publish the peer review history of their article (what does this mean?). If published, this will include your full peer review and any attached files.

Reviewer #1: No

Reviewer #2: No

Reviewer #3: No
---

## [Decision Letter · Decision Letter 1]

1 Mar 2021

Dear Dr. Weiser,

We are pleased to inform you that your manuscript 'Neuraminidase B controls neuraminidase A-dependent mucus production and evasion' has been provisionally accepted for publication in PLOS Pathogens.

Best regards,

Carlos Javier Orihuela, PhD

Associate Editor

PLOS Pathogens

Michael Wessels

Section Editor

PLOS Pathogens

Kasturi Haldar

Editor-in-Chief

PLOS Pathogens

orcid.org/0000-0001-5065-158X

Michael Malim

Editor-in-Chief

PLOS Pathogens

orcid.org/0000-0002-7699-2064

Reviewer Comments (if any, and for reference):

Reviewer's Responses to Questions

**Part I - Summary**

Reviewer #1: (No Response)

Reviewer #3: This is an interesting and well written manuscript that addresses the long-standing question of what the biological function of NanB might be, given that all previously known contributions of pneumococcal neuraminidase are attributed to NanA. The experiments and data are logical and support the conclusions of the Authors. The Authors' revisions have successfully addressed my concerns.

**Part II – Major Issues: Key Experiments Required for Acceptance**

Reviewer #1: (No Response)

Reviewer #3: None

**Part III – Minor Issues: Editorial and Data Presentation Modifications**

Reviewer #1: (No Response)

Reviewer #3: None

PLOS authors have the option to publish the peer review history of their article (what does this mean?). If published, this will include your full peer review and any attached files.

Reviewer #1: No

Reviewer #3: No

---

## [Editor Report · Acceptance letter]

30 Mar 2021

Dear Dr. Weiser,

We are delighted to inform you that your manuscript, "Neuraminidase B controls neuraminidase A-dependent mucus production and evasion," has been formally accepted for publication in PLOS Pathogens.

Best regards,

Kasturi Haldar

Editor-in-Chief

PLOS Pathogens

orcid.org/0000-0001-5065-158X

Michael Malim

Editor-in-Chief

PLOS Pathogens

orcid.org/0000-0002-7699-2064